# Salt Tolerant *Bacillus* Strains Improve Plant Growth Traits and Regulation of Phytohormones in Wheat under Salinity Stress

**DOI:** 10.3390/plants11202769

**Published:** 2022-10-19

**Authors:** Muhammad Ayaz, Qurban Ali, Qifan Jiang, Ruoyi Wang, Zhengqi Wang, Guangyuan Mu, Sabaz Ali Khan, Abdur Rashid Khan, Hakim Manghwar, Huijun Wu, Xuewen Gao, Qin Gu

**Affiliations:** 1Key Laboratory of Integrated Management of Crop Diseases and Pests, Ministry of Education, Department of Plant Pathology, College of Plant Protection, Nanjing Agricultural University, Nanjing 210095, China; 2Shenzhen Batian Ecological Engineering Co., Ltd., Shenzhen 518057, China; 3Biotechnology Department, College of Environmental Sciences, COMSATS, Abbottabad 22060, Pakistan; 4Lushan Botanical Garden, Chinese Academy of Sciences, Jiujiang 332000, China

**Keywords:** genetic features, halophilic *Bacillus* spp., phytohormones, ROS regulation, salinity, wheat

## Abstract

Soil salinity is a major constraint adversely affecting agricultural crops including wheat worldwide. The use of plant growth promoting rhizobacteria (PGPR) to alleviate salt stress in crops has attracted the focus of many researchers due to its safe and eco-friendly nature. The current study aimed to study the genetic potential of high halophilic *Bacillus* strains, isolated from the rhizosphere in the extreme environment of the Qinghai–Tibetan plateau region of China, to reduce salt stress in wheat plants. The genetic analysis of high halophilic strains, NMCN1, LLCG23, and moderate halophilic stain, FZB42, revealed their key genetic features that play an important role in salt stress, osmotic regulation, signal transduction and membrane transport. Consequently, the expression of predicted salt stress-related genes were upregulated in the halophilic strains upon NaCl treatments 10, 16 and 18%, as compared with control. The halophilic strains also induced a stress response in wheat plants through the regulation of lipid peroxidation, abscisic acid and proline in a very efficient manner. Furthermore, NMCN1 and LLCG23 significantly enhanced wheat growth parameters in terms of physiological traits, i.e., fresh weight 31.2% and 29.7%, dry weight 28.6% and 27.3%, shoot length 34.2% and 31.3% and root length 32.4% and 30.2%, respectively, as compared to control plants under high NaCl concentration (200 mmol). The *Bacillus* strains NMCN1 and LLCG23 efficiently modulated phytohormones, leading to the substantial enhancement of plant tolerance towards salt stress. Therefore, we concluded that NMCN1 and LLCG23 contain a plethora of genetic features enabling them to combat with salt stress, which could be widely used in different bio-formulations to obtain high crop production in saline conditions.

## 1. Introduction

Wheat (*Triticum aestivum* L.) is grown on 17% of cultivated land worldwide [1]. It is considered to be an important cereal crop that provides 82% of protein and 85% of calories to the world population. It is an essential staple food for approximately 41 countries, but it can be cultivated worldwide due to its unique adaptability nature [2]. Staple food demand is increasing day by day, so it is necessary to produce wheat in abundance to fulfil the needs of the current increasing population throughout the globe [3]. Wheat is reported to be adversely affected by abiotic stresses worldwide [4]. Among abiotic stresses, soil salinity is known to be a major constraint in wheat production that results in terrible yield loss [5]. Salinity has a detrimental effect on wheat growth by inducing physiological and metabolic disorders that lead to oxidative stress, osmotic stress, nutritional abnormalities, membrane dysfunction, reduced photosynthetic activity and improper hormone function [6,7]. Plants under salt stress often overproduce reactive oxygen species (ROS), i.e., superoxide (O_2_^−^) and hydrogen peroxide (H_2_O_2_) leads to protein, cell wall and nucleic acid damage [8,9]. Salinity has become more notorious than ever before due to global warming, along with possibly undesirable effects on the economy, such as in agriculture and food security, mainly in developing nations [10]. Around 300 million hectares of cultivated irrigated land is affected by salinity, and half of it lies in Pakistan, India, China and USA [11].

The application of Plant Growth Promoting Rhizobacteria (PGPR) has been documented to reduce abiotic stresses in plants that will enhance sustainable agriculture [12,13]. The use of PGPR to enhance plant growth under salt stress is an emerging technology [14]. Many researchers are working on the potential of PGPR for salt stress alleviation in agricultural crops [15]. The cross-talk between microbes and plants provides multifaceted mechanisms that induce salt stress resistance in plants [16]. Previous studies revealed that PGPRs such as *Pesudomonas* spp. and *Bacillus* spp. isolated from saline soil can promote plant growth under saline conditions [6,17]. This idea has attracted the attention of researchers to discover efficient biotechnological approaches involving the application of PGPR as a bio-resource to reduce salinity effects and enhance plant growth under salt stress [17,18]. Recently, many significant results have been achieved through the application of PGPR for salt stress reduction in wheat plants, leading to enhanced growth and productivity in the saline area [16]. PGPR or certain compounds can help to withstand salt stress by modulating hormonal, photosynthetic and ROS scavenging pathways in plants [8,10]. Among PGPR, *Bacillus* strains have attracted significant focus because of their important characteristics such as gram-positivity, ability to make spores, colonise roots and promote plant growth [19,20]. *Bacillus* is an important and well-known group of microbes that can produce multiple products with the potential to show resistance to unfavourable environmental conditions, due to their natural rigidness and stability [21,22]. *Bacillus* strains have been used to alleviate abiotic stresses in plants, however, the genetic features enabling them to show resistance were rarely studied under extreme conditions [20,23]. Therefore, it is important to develop high salinity-tolerant *Bacillus* strains with salt-resistant genetic features, which simulate the production of bioactive osmotic compatible solutes that cope with hostile salt stress conditions [24].

The expression analysis of key genetic features in the genome of *Bacillus* enable them to show tolerance under stress conditions [25]. The genes that belong to different families, i.e., membrane transport, osmotic regulation, signal transduction and oxidative stress and antioxidant enzymes are mainly attributed to the potential of *Bacillus* strains to endure salt stress [25,26,27]. The genes expressed under salt stress result in certain transcriptional changes to improve membrane function, antioxidant enzyme activity, and water maintenance inside the cell through regulating various signalling pathways [28]. In addition to these essential genetic features, the formation of biofilm by *Bacillus* is an important property to colonise and produce secondary metabolites under adverse conditions [29].

The stress signals in plants, such as proline accumulation, lipid peroxidation and abscisic acid, can regulate stress in plants, as these indigenous signal molecules can stimulate various plants biological processes both distantly and locally [30]. Few PGPRs, including *Bacillus*, have the potential to regulate these stress signalling molecules and lead to stress alleviation in plants [15]. Under salt stress conditions, abscisic acid (ABA), which promotes root growth and regulates water balance in plants [31], is also reported to be downregulated by the application of PGPRs [32]. Malondialdehyde (MDA) is a well-known marker for oxidative stress injury and membrane lipid peroxidation under stressful conditions in plants [33]. Interestingly, PRGR was also able to decrease the level of MDA in plants [9]. Similarly, the use of PGPR could positively modulate the activity of Proline, which is an osmoprotectant which plays a vital role in plant defence against salt stress [34,35]. The *Bacillus* strains can also regulate the plant stress response by regulating phytohormones in plants [36]. The presence of the ACC deaminase gene in *Bacillus* can lower ethylene concentration under abiotic stresses that results in plant growth promotion [37]. The upregulation of phytohormones such as cytokinin, gibberellin, and auxin by *Bacillus* under stress conditions in plants results in wheat plant growth promotion [38]. The main plant cell wall regulator expansin is reported to be stimulated by PGPR under stress conditions that results in plant growth promotion [16].

Therefore, the aim of the present study was to evaluate the potential of *Bacillus* strains isolated from the Qinghai–Tibet region of China to enhance plant growth and reduce the adverse effect of saline stress on wheat. The genetic screening and expression analysis revealed *Bacillus* spp. key genetic features that make them capable of resistance under salt stress conditions by regulating various metabolic processes. The halophilic *Bacillus* strains (salt-loving) were found to have positive effects on wheat growth parameters by minimizing the adverse effect of salinity.

## 2. Results

### 2.1. Bacterial Growth and Biofilm Potential under Various Salt Concentrations

The selected strains and FZB42 were able to grow on up to 10% NaCl LB medium. However, two strains, i.e., NMCN1 and LLCG23, were further able to grow on up to 18% NaCl and 14% NaCl, respectively, and showed high resistance under salt stress conditions. NMCN1 and LLCG23 were then grown in LB liquid medium with different NaCl concentrations (1–18%) and the growth pattern was observed for each strain. Significant growth was observed for NMCN1 at up to 16% NaCl, followed by LLCG23 up to 14% NaCl medium. Furthermore, less growth of NMCN1 on 18% NaCl was observed than on 16% NaCl medium, as shown in Figure 1A,B.

Biofilm formation potential results of *Bacillus* strains (NMCN1, LLCG23 and FZB42) under different NaCl concentration (1%, 4%, 8%, 10%, 16%, and 18%) at 37 °C for 5 days showed that all the strains were able to form the finest biofilm up to at 8% however the potential of FZB42 to form biofilm at 10% was reduced and stopped at high NaCl concentrations of 14–18%. The NMCN1 was observed as the best strain to form biofilm continuously up to on 18% NaCl followed by LLCG23. The weak biofilm was noticed at 18% NaCl for NMCN1 while LLCG23 was unable to grow at the same conditions as shown in Figure 1C.

### 2.2. Relative Expression Profiling of Predicted Genes through qPCR

The signal transduction genes, i.e., *DegS* and *DegU*, through which bacteria can perceive and respond to abiotic stress were detected in the selected strains. The gene superoxide dismutase *SodA* plays an important role to minimise the effect of free radicals produced in bacteria under stress conditions detected in selected strains. The *OhrR* gene was also predicted in the selected strains that plays an important role in the regulation of hydrogen peroxide in the bacterial cell under salt stress. The *OstB* gene, which regulated osmotic potential in the bacteria and plays an important role in salt stress, was detected in the selected strains. The glycine betaine genes (*OPuD* and *OpuAC*) are also responsible for osmotic regulation, and were detected in selected strains. The expression profile of selected genes involved in salt resistance was studied in NMCN1 and LLCG23 in comparison to FZB42 grown in different NaCl concentrations (1%, 10%, 14% and 16%) for four days. The results showed a linear upsurge in the expression of salt-resistant genes in NMCN1 and LLCG23, as compared with control. The expression of salt-resistant genes in FZB42 was also stimulated up to 10% NaCl, however, with increase in saline conditions, i.e., 14% NaCl and 16% NaCl, FZB42 was unable to stimulate the expression of salt-resistant genes. The high expression of the selected salt resistant genes was noticed in NMNC1 in linear order with increased saline conditions, followed by LLCG23. The gene involved in signal transduction, i.e., *DegU* showed high expression in saline conditions (16% NaCl) as compared with LLCG23, which might impart efficient sensing potential to the NMCN1 and makes it able to induce high resistance to salt stress. The *SodA* gene was noticed to be highly expressed in bacteria at 16% NaCl saline condition followed by LLCG23 to regulate ROS inside the strains. The *OpuAC* gene responsible for glycine betaine involved in osmotic protection and regulation also showed increased expression in NMCN1, followed by LLCG23 under 16% NaCl concentration. The *OstB* and *OhrR* genes, which are reported to play an important role in osmotic regulation under high saline conditions, were also observed to be highly overexpressed in NMCN1. The *ComA* genes involved in quorum sensing regulation were also noticed to be highly stimulated in NMCN1 and LLCG23. The overall results showed that all the selected salt-resistant genes showed an upsurge in the expression level with increasing saline conditions in the selected strains as shown in Figure 2.

### 2.3. ROS Regulation Potential of Bacillus Strains under Salt Conditions

*Bacillus* strains cultured in 16% NaCl LB media for four days were studied for ROS regulation through the staining method. The bacterial cells stained with dye for ROS detection were observed under microscope. The results showed that the salt-resistant strains NMCN1 and LLCG23 maintained lower levels of ROS grown for four days in 16% NaCl LB media at 37 °C, whereas the FZB42 strain was observed with a significant increase in ROS level grown in the same conditions, as shown in Figure 3.

### 2.4. Seedling Growth and Root Morphological Parameters

Wheat seedlings’ growth along with different root morphological parameters were noticed to be enhanced by *Bacillus* strains under different NaCl concentrations (0 mmol–200 mmol). The FZB42 was observed to have high growth-promoting potential under normal conditions (0 mmol) as compared to NMCN1 and LLCG23, however, a decrease in growth-promoting potential for FZB42 was observed at a high saline condition of 200 mmol. The halophilic strain NMCN1 showed maximum growth potential under 200 mmol, followed by LLCG23 and FZB42. The vigour index (VI) calculated from the data of total seedlings’ length and percent germination was noticed to be maximum under high NaCl concentration (200 mmol) in wheat seedlings inoculated with NMCN1 (VI = 1289), found to be significantly higher than control seedlings and the seedlings inoculated with LLCG23 and FZB42. The second highest VI was observed for LLCG23 grown under 200 mmol NaCl stress. Moreover, under normal conditions (0 mmol) the best result for FZB42 strain was noticed in growth promotion and vigour index of wheat seedlings having V1 value (1700) higher than un-inoculated control seedlings VI as shown in Figure 4.

Moreover, the data related to morphological root parameters such as root diameter, root volume, root length, root tips and root surface area were observed to be increased for NMCN1-inoculated wheat seedlings, and several tips were found to be increased in the case of seedlings inoculated with NMCN1, followed by LLCG23 and FZB42 under high stress conditions (200 mmol NaCl) in comparison with non-treated seedlings grown in the same conditions. The overall data demonstrated that wheat seedlings treated with selected *Bacillus* strains have increased root morphological parameters as compared with control under normal conditions (200 mmol), whereas NMCN1 and LLCG23 were found to be more efficient to alleviate the risk of high salt stress conditions (200 mmol) as compared with FZB42 as shown in Figure 5.

### 2.5. Plant Stress Response Parameters Quantification and Expression Analysis

Plant stress response parameters were observed to be regulated by *Bacillus* strains. The UPLC results for ABA level showed significant reduction for NMCN1-inoculated wheat plants followed by LLCG23 under high saline conditions (200 mmol NaCl) in comparison with non-inoculated plants as control. The ABA level in wheat plants inoculated with NMCN1 was 560 ng/g FW and 640 ng/g FW for LLCG23 in comparison with 1200 ng/g FW ABA level in control plants. Under normal conditions, FZB42 was observed to reduce ABA level significantly, i.e., 432 ng/g FW compared with 750 ng/g FW of ABA level in control. The wheat plant inoculated with NMCN1 and LLCG23 also showed reduced ABA level under normal conditions as shown in Figure 6.

The Malondialdehyde (MDA) level was measured through the spectrophotometric method in inoculated and un-treated *Bacillus* strain wheat plants under saline conditions (200 mmol NaCl). In inoculated wheat plants the MDA level was noticed to be significantly reduced, i.e., 0.493 µmole MDA/g FW for NMCN1 and 0.544 µmole MDA/g FW for LLCG23 in comparison with control MDA content of 0.897 µmole MDA/g FW under salt stress, as shown in Figure 6. The MDA content of wheat plants treated with FZB42 under salt stress conditions (200 mmol) was 0.797 µmole MDA/g FW that is slightly high in comparison with halophilic strains (NMCN1 and LLCG23). Under normal conditions, measured MDA content for all treatments showed no significant increase when compared with control.

In the present study proline content in wheat plants was noticed to be significantly increased in wheat seedlings inoculated with NMCN1 and LLCG23 under salt stress conditions (200 mmol NaCl). The inoculated wheat plants showed a maximum proline level, i.e., 1.272 µg/g FW and 1.145 µg/g FW, respectively, in comparison with control 0.293 µg/g FW under saline condition. The measured proline content for all treatments under normal conditions (0 mmol) showed no significant increase as compared to their control. The candidate genes involved in the different stress response parameters were observed to be highly regulated by halophilic *Bacillus* strains. The keys genes involved in the MDA and ABA pathways were observed to be significantly downregulated in NMCN1 and LLCG23 treated wheat plants under high saline conditions. Moreover, the proline encoding *P5CS* was observed to be upregulated in the plants treated with the selected strains. The maximum increase in the expression of *P5CS* in plants under saline condition was noticed for NMCN1, followed by LLCG23. The results showed that stress-responsive genes are significantly regulated in wheat plants by salt resistance strains (MNCN1 and LLCG23) as shown in Figure 6.

### 2.6. Bacillus Strains Enhance Plant Growth under Saline Conditions

In the current study plant growth and root morphological parameters were noticed to be significantly increased in wheat plant inoculated with bacteria under salt stress as shown in Figure 7 and Figure 8. The high halophilic strain NMCN1 followed by LLCG23 were observed for significant improvement in wheat plant growth, one-fold increase in shoot/root length and fresh/dry weight under salt stress (200 mmol NaCl) whereas the strain FZB42 slightly enhanced plant growth under the same condition. Furthermore, all the *Bacillus* strains were noticed for nearly equal enhancement in wheat pants under normal conditions. An increase in salt stress conditions, i.e., 100 mmol–200 mmol NaCl concentrations negatively affected the growth of un-inoculated wheat plants however the effect was reduced when the plants were inoculated with halophilic strains. The NMCN1 was observed to highly alleviate the negative effect of increased saline condition. Overall salt stress was observed to be alleviated in the presence of halophilic strains, i.e., NMCN1 and LLLCG23 in wheat plants.

### 2.7. ROS Reduction in Wheat Roots by Bacillus Strains under Salt Conditions

The halophilic strain NMCN1 was observed to highly alleviate ROS production in wheat roots grown under the high salt condition as compared to FZB42. The strains NMCN1 played an efficient role in ROS reduction in wheat roots and thus might reduce the severe effect of salt stress on wheat roots, as shown in Figure 9.

### 2.8. Regulation of Photosynthesis in Wheat by BACILLUS Strains

Wheat plants inoculated with *Bacillus* strains under salt stress conditions were studied for photosynthesis and stomatal conductance regulation. The current findings showed that all the strains were able to regulate the photosynthesis and stomatal conductance in inoculated and un-inoculated wheat plants under saline or normal conditions. The NMCN1 strain was noticed to significantly improve the photosynthesis rate and stomatal conductance in wheat plants under high saline conditions (200 mmol NaCl). The second important results under salt stress for wheat photosynthetic and stomatal conductance enhancement were noticed for LLCG23 strain. *Bacillus* strain FZB42 was also observed for the most significant improvement in the photosynthetic activity, as well as stomatal conductance compared with control plants under normal conditions, as shown in Figure 10.

### 2.9. Expression Analysis of Growth and Salt Related Genes in Wheat

The expression of genes related to growth promotion and salt resistance in wheat plants under saline conditions were analysed in the presence of *Bacillus* strains (NMCN1, LLCG23 and FZB42). The current results showed that all the strains have the potential to efficiently regulate the growth promoting and salt resistance genes in wheat plants under saline conditions (200 mmol NaCl). The most significant results were obtained for NMCN1 which possessed high potential to upregulate the genes involved in growth promotion and salt tolerance in wheat plants. The plants inoculated with NMCN1 under salt stress (200 mmol NaCl) significantly overexpressed the genes related to expansin (*expA1*), cytokinin (*CKX2*) and auxin (*ARF*), followed LLCG23 and FZB42. The expression of ethylene encoding gene (*ERF*) was noticed to be highly downregulated in wheat plants treated with NMCN1 strain grown under the same stress condition. Under normal conditions the plants inoculated with FZB42 were noticed for high upregulation of growth-related genes. Overall high upregulation in the expression of growth promoting genes (*CKX2*, *expA1* and *ARF*) was observed, while significant down-regulation was observed for ethylene encoding gene (*ERF)* in wheat plants treated with high halophilic strains, NMCN1 and LLCG23, under the saline conditions (200 mmol NaCl), as shown in Figure 11. The expression of salt stress-related genes in wheat plants inoculated with *Bacillus* strains was observed to be significantly stimulated under saline conditions, i.e., (200 mmol NaCl). The wheat plants treated with highly halophilic bacteria, NMCN1, were noticed to highly over-express the salt-resistant genes (*MYB, DREB2*, *HKT1* and *WRKY17*), followed by LLCG23 and FZB42. The NMCN1 was found to highly upregulate the expression of selected genes with 4–5-fold increase in plants grown under high salt stress as compared with control. The FZB42 was also observed to increase the expression of the selected genes but less than NMCN1 and LLCG23 as shown in Figure 11.

## 3. Discussion

The plant faces many challenges due to its sessile nature in the form of abiotic and biotic stresses [39,40]. Soil salinity is considered as a major constraint among abiotic stresses that reduces crop productivity worldwide [40,41]. Through molecular breeding and genetic engineering plants’ defence mechanisms have been improved for the purpose of alleviating the adverse effects of abiotic stresses on important agricultural crops [6,24], but these methods are mostly laborious and time-consuming [10,20]. Therefore, the use of PGPR is considered as a fast and eco-friendly approach among the researchers to reduce salt stress in plants [18,42]. Plants’ abiotic stress tolerance mechanism can be improved by inoculating them with PGPR [2,43]. In the present study the *Bacillus* spp. strains, i.e., NMCN1 and LLCG23 isolated from the extreme environment of Qinghai–Tibetan region of China showed high salinity stress tolerance and were able to grow at up to 16% and 18% NaCl concentrations. The reference *Bacillus velezensis* FZB42 strain was also observed to grow on LB medium under 10% NaCl concentration.

In the present study, the halophilic strains were explored for their genetic features that could take part in alleviating salt stress condition. The results showed that key genes responsible for membrane transport, osmotic regulation, regulation of reactive oxygen species, cell signalling and fatty acid and lipid metabolism that enable the bacteria to combat with salt stress were mostly detected in NMCN1 and LLCG23, as compared with FZB42. All these genetic features (*SodA, OpuAC, OPuD*, *OstB, ComA* and *DegU)* previously reported in the literature were known to play an important role in abiotic stress (Cold and Salt) resilience [17,20]. The whole genome genetic analysis of the halophilic strains (NMCN1 and LLCG23) also showed that the selected strains possess a plethora of genes to combat high salinity stress. The gene *SodA* linked with ROS regulation was noticed to be present in the selected strains. The presence of the ROS regulation genes help *Bacillus* strains to reduce the adverse effects of oxidative stress induced after salt stress. Moreover, the most important signal transduction genes *DegS* and *DegU* were also detected in halophilic strains. The predicted genes in halophilic strains linked with signal transduction suggested that our bacteria have the ability to sense different stresses that enable the strains to show response to such environmental shocks at early stage. The genes related to osmotic stress regulation, i.e., *OpuAC, OPuD* and *OstB* were also predicted in salt-resistant *Bacillus* strains that play an important role in glycine betaine synthesis which is considered as an important osmoprotectant [44]. The *ComA* gene was also found in the halophilic strains which is an important regulator of quorum-sensing previously reported in the literature.

The upregulation in the expression of salt-resistant genes found in the halophilic strains (NMCN1 and LLCG23) reveals their potential to survive under high saline conditions. The FZB42 was unable to survive under high salt stress conditions due to lower expression of selected genes involved in salt stress tolerance. Moreover, the high expression of quorum-sensing regulator gene (*ComA*) was for the first time noticed in the halophilic strains that could help them to survive under saline conditions, however, it was previously reported as an important transcriptional activator of other physiological responses in bacteria [45]. The genes involved in the signal transduction pathway, i.e., *DesS* and *DegU,* were also observed to be highly expressed in NMCN1 and LLCG23, showing their potential to sense and respond rapidly to the changing environmental stresses and thus support the previous findings that documented the main role of these genes in *Bacillus Pumilus* 7P strain under salt stress. The high upsurge in the expression of the signal transduction genes of NMCN1 and LLCG23 reveal that the strains were able to perceive salt stress signals in an effective manner as compared to FZB42, which was unable to grow under high saline conditions due to lower expression of the mentioned genes. The genes, i.e., *OhrR* and *OpuAC,* play an important role in osmotic regulation and were also noticed to be over-expressed in NMCN1 and LLCG23 under high salt stress conditions, and thus support the previous findings related to the activation of these transcription factors upon abiotic stress [46]. The genetic screening and expression analysis of the predicted genes involved in abiotic stress resistance suggested that the halophilic *Bacillus* strains under study contain several genetic features supporting them to survive under saline stress condition.

The Reactive Oxygen Species (ROS) produced in bacteria under salt–stress conditions disrupt the cellular function that leads to cell death [33,47]. In the present study the halophilic strains (NMCN1 and LLCG23) were found to have less ROS as compared with FZB42, indicating that these strains have a high potential to regulate oxidative stress for the purpose to well survive under salt stress condition. Furthermore, biofilm formation is another important characteristic of *Bacillus* strains that help them endure environmental stresses. The growth of bacteria in the self-secreted matrix for biofilm formation help them survive under adverse conditions [48]. Previous study showed the distortion in bacterial biofilm structures under high salt stress conditions [49], but little evidence is reported about the development of biofilm under high saline conditions. The current research first time showed thick and fine biofilm formation by halophilic strains (NMCN1 and LLCG23) for the purpose to survive under high saline conditions. Salinity affected plant growth, inducing reductions in the roots system [50]. We assume that the selected halophilic *Bacillus* stains possess the unique potential of biofilm formation and producing important metabolites under salt stress. The biofilm formation of halophilic PGRR can be linked with their ability to colonize plant roots in order to alleviate the adverse effects of salt stress on plants.

The halophilic strains (NMCN1 and LLCG23) were further noticed for their important role in reducing the adverse effects of salinity on wheat plants grown under saline conditions. The strains have the potential to regulate stress responsive parameters, such as proline accumulation, lipid per-oxidation and abscisic acid (ABA) in salt stress challenged wheat plants. The halophilic strains were observed for efficiently, regulating the mentioned parameters as compared with FZB42 strain. The ABA level was noticed to be high in plants grown under saline conditions but wheat plants inoculated with halophilic strains were observed to have decreased ABA levels for the purpose of mitigating salinity stress that supports the previous findings of ABA level reduction in inoculated *Bacillus* strains plants facing salt stress [51]. Furthermore, the accumulation of proline and reduction in malondialdehyde (MDA) levels are important indicators in plants to combat various environmental stresses, as proline is an important osmoprotectant [52], whereas the decreased level of MDA indicated less membrane damage of plants inoculated with halophilic *Bacillus* strains under saline conditions [53]. The halophilic strains in the current study were also found to regulate proline and MDA levels in a very efficient manner in wheat plants grown under salt stress conditions. The results showed an increased accumulation of proline and decreased MDA level in inoculated halophilic bacteria wheat plants grown under saline conditions. Moreover, expression of the key genes responsible for these stress responsive parameters were also noticed to have the same pattern that further supports our claim of positive regulation for alleviating salt stress in bacteria inoculated plants. Besides regulation of stress responses, the current study also reported the important role of halophilic *Bacillus* strains (NMCN1 and LLCG23) in the positive regulation of the photosynthesis rate and stomatal conductance of wheat plants under salt stress. There have been some studies suggesting the improvement in these parameters by inoculated bacteria in normal conditions [54] but no previous study has indicated such results under salt stress in wheat plants.

The use of PGPR to promote plant growth has been previously well documented in the literature [55]. There are many studies related to plant growth promotion by PGPR under drought stress conditions [56]. Similarly, the application of *Pseudomonas* spp. enhances the growth of pea plant under drought stress [57]. We hereby report the halophilic strains (NMCN1 and LLCG23) for their major role in enhancing wheat growth through seed inoculation under saline conditions. Furthermore, all the selected *Bacillus* strains (NMCN1, LLCG23 and FZB42) were observed to have a high potential of wheat plant growth promotion under normal conditions, however, under salt stress conditions (200 mmol) high reduction in plant growth promoting potential of FZB42 strain was observed for the first time as compared to other halophilic strains NMCN1 and LLCG23. Previously, *Bacillus* and *Pseudomonas* strains were studied for their positive effects on seed germination, vigour index and root morphology of young growing plants [58,59], but in the current study significant improvements in vigour index and root morphological parameters of wheat seedlings were first time reported through inoculation of highly salt-tolerant *Bacillus* spp. strains NMCN1and LLCG23. The growth parameters such as fresh/dry biomass, root and shoot length of wheat inoculated with the selected bacteria were noticed to be significantly increased as compared to control plants.

The genes related to phytohormones, i.e., cytokinin (*CKX2)*, auxin (*ARF*) and expansin (*exPA1*) were observed to be upregulated while ethylene gene (*ERF*) was noticed to be downregulated in wheat plants inoculated with selected *Bacillus* strains under salt stress conditions, as there are many studies elaborating the importance of phytohormones in plant growth promotion [24,60]. Previous studies also showed upregulation of phytohormones, i.e., cytokinin, auxin and expansin related genes and downregulation of the ethylene gene in abiotic stress plants inoculated with PGPR [20,61]. The regulation of genes related to phytohormones in the same pattern by halophilic *Bacillus* strains NMCN1 and LLCG23 significantly showed their main role in wheat growth promotion under salt stress conditions. Thus, we noticed that wheat plant growth promotion through inoculated bacteria under saline conditions is an accumulative effect of their ability to regulate phytohormones and salt resistance genes. Moreover, few detailed studies related to salt stress reduction in wheat plants by halophilic *Bacillus* strains through regulation of salt-resistant genes (*MYB, DREB2*, *HKT1* and *WRKY17*) were reported in the literature [20,40]. The present study evaluated the potential of *Bacillus* strains to enhance plant growth and reduced the adverse effect of saline stress on wheat through regulation of salt resistant genes. Furthermore, these halophilic *Bacillus* strains (salt-loving) were found to have positive effects on wheat growth parameters and minimize the adverse effects of saline conditions.

## 4. Materials and Methods

### 4.1. Growth and Biofilm Formation of Bacillus Strains

*Bacillus* strains were obtained from Laboratory of Biocontrol and Bacterial Molecular Biology, Nanjing Agricultural University. These strains were isolated from Qinghai–Tibet plateau region of China. These strains were screened for their growth and biofilm formation on Luria-Bertani (LB) agar plates under different salt (NaCl) concentrations (1–18%) at 37 °C for 4 days along with *Bacillus velezensis* FZB42, a model plant growth promoting and bio-control strain as a control. Two strains (NMCN1 and LLCG23) that showed high salt resistance were then grown in liquid culture to study their growth pattern at optical density OD_600_ for different time intervals (maximum up to 96 h) at 37 °C through a spectrophotometer. The growth pattern of each strain was determined by making a growth curve for the different time intervals. Furthermore, to know the effects of different salt concentrations on the biofilm formation, the selected strains were grown on LB liquid culture. The selected salt-resistant strains were grown in a 20 mL flask in LB at 37 °C till 1.0 OD_600_ and then 4 μL of each bacterial culture was mixed with LB with different salt concentrations in 12 wells sterile costar^®^ culture plates. After sealing the plates were kept at 37 °C for 5 days. The experiment was repeated thrice with three replicates for each treatment.

### 4.2. Genetic Analysis of Bacillus Strains and Expression of Predicted Salt Related Genes

Each strain was screened for predicted salt tolerance genes in its genome using the bioinformatics tool RAST. The predicted genes were amplified through PCR with primers designed through the Primer Quest tool and detected on gel electrophoresis, as previously described [20]. Total RNA was isolated from each PGPR, i.e., NMCN1, LLCG23 and FZB42, grown under different NaCl concentrations (1%, 10% 14% and 16%) in liquid LB culture at 37 °C for 4 days in a shaker incubator. The samples were harvested after 4 days post-inoculation for the expression pattern of the selected salt resistance genes. The RNA from each sample was extracted through (Bio-tek-OMEGA, Inc., Norcross, GA, USA) kit. The purity and concentration of extracted RNA was calculated using Nanodrop (Nanodrop 1000, Thermo Scientific, Wilmington, DE, USA) at an absorbance of 260/280 nm. The RNA sample of each strain was then subjected to cDNA synthesis using a 5× All-In-One RT Master Mix (with AccuRT Genomic DNA Removal Kit) by Applied Biological Materials Inc. (abm^®^, Beijing, China), according to the given protocol.

The expression profile of each salt resistant gene in the selected strain was observed by RT-qPCR performed in a Step One Real-Time PCR System (Applied BioSystems, Foster City, CA, USA) machine based on the modulation in fluorescence relative to the cyclic increase in PCR products. The Syber Green (Takara Bio, Beijing, China) was used as a detector that emits fluorescence while binding to cDNA. The fluorescence was recorded with threshold cycle ©. The PrimerQuest tool of IDT was used to design primers for the selected salt resistant genes listed in Appendix A. The *rpsJ* genes, previously mentioned as a housekeeping gene in *Bacillus* strains, were used in this current study [20,62]. The 20 µL volume of PCR mixture contained 10 µL 2X SYBR premix Ex Taq (Takara Bio, Beijing, China) (TilRNaseH Plus) with reference dye ROX (0.4 µL), 0.4 µL forward and reverse primers (20 mmol), 2 µL cDNA (100 ng) and 6.8 µL ddH_2_O. The PCR was programmed as: 95 °C for 30 secs as initial denaturation temperature, 40 cycles of 95 °C for 5 secs and 34 secs at 60 °C. The T melt curve was observed to find out the accuracy of the reactions at the end of the PCR process. At the end comparative C method of 2^−ΔΔCT^ reported by [63] was used to analyse the data.

### 4.3. ROS Detection in the Selected Strains Grown under Saline Conditions

The severity of cell damage or dysfunction is closely related to the production of ROS under unfavourable conditions [64]. As an indicator of cell damage or cell disturbance, ROS was detected in each strain grown under different salt concentrations. For ROS detection NMCN1, LLCG23 and FZB42 strains were grown on LB medium with different salt concentrations and incubated at 37 °C for 4 days. The cells of each strain were harvested after 4 days of inoculation in 1.5 mL Eppendorf tube through centrifugation at 1000 rpm. The harvested cells were incubated at 25 °C for 30 min in a dichloro-dihydro-fluorescein diacetate (DCFH-DA) (JianCheng Bioengineering, Nanjing, China [20,65] and 10 mM sodium phosphate buffer with 7.4 pH. The damage *Bacillus* cells containing ROS was stained with chemical dye and the fluorescence was detected with microscope Olympus1 × 71 and Image-Pro express software v.6.2 (Olympus, Tokyo, Japan).

### 4.4. Determination of Salt Stress Effects on Vigour Index of Wheat Seedlings

The effect of salt stress on wheat germination and seedlings growth was observed through vigour index (VI) measurement. Surface sterilisation of wheat seeds was performed with 5% sodium hypochlorite and then washed with 70% ethanol and autoclaved water. The sterilised seeds were inoculated with high salt-resistant strains i.e., NMCN1 and LLC23 and well-known high salt-sensitive plant growth-promoting bacteria FZB42. The seeds were transferred to sterilised 9 cm Petri plates with 0. 3% water agar of different NaCl concentrations (0 mmol, 100 mmol, 150 mmol, and 200 mmol) sealed with parafilm and incubated at 25 °C in dark. The seeds simply dipped in water were used as control and were placed on different salt concentrations of 0.3% water agar. The germination was observed when the radicals were half of the shoot length. Five germinated seedlings were then transferred from each treatment to Petri plates containing water agar (0.3%) of different salt concentrations and allow them to grow for 7 days at 25 °C to find the effects of salt stress on seedlings stage in the presence of the selected strains. The percent germination and VI were calculated by the formula previously reported [58].

### 4.5. Measuring Effects of Salt Stress on Seedling Roots

The ability of the PGPR inoculated wheat seedlings was determined to grow under different salt concentrations of 0.3% agar water for 7 days at 25 °C in sealed petri plates (9 cm). Briefly, the seeds were surface sterilised and inoculated with different PGPRs i.e., MNCN1, LLG23, and FZB42. The seeds without inoculation were used as control. The seeds were then allowed to germinate and transferred to petri plates (9 cm) having different salt concentrations of 0.3% water agar and allowed to grow for 7 days at 25 °C. To study the root morphology three seedlings were then taken from each treatment. The root length, surface area, root diameter, root volume, and the number of each root seedling tips were calculated through Rhizoscanner (EPSON Perfection V700 Photo, Epson America, Long Beach, CA, USA), equipped with WinRHIZO software offered by Regent Instruments Co., (Sainte-Foy, Quebec, QC, Canada) [58].

### 4.6. Salt Stress Alleviation and Plant Growth Promotion by Bacillus Strains

Seven days’ wheat seedlings were transplanted into pots containing sterilised soil of equal size placed in a greenhouse. The NMCN1, LLCG23 and FZB42 strains were grown in the LB medium for 24 h at 37 °C. The cells were recovered by centrifugation and resuspended in the sterilised water until the final OD_600_ of 1.0 (1 × 107 CFU/mL). After one day of inoculation, 20 mL NaCl solutions of different concentrations (0 mmol, 100 mmol, 150 mmol and 200 mmol) were added to the inoculated and non-inoculated wheat plants. The non-inoculated plants grown with simple water or in the presence of different salt concentrations were used as control. Each treatment was performed in triplicate with three wheat seedlings in each pot. After 9 days’ post-inoculation, the effects of salt stress were recorded. The growth parameters, i.e., biomass were measured and the data were used for indication of growth promotion.

### 4.7. Rhizoscanning of Wheat Plant Roots for Morphological Traits

The root morphological study is mostly carried out in PGPRs-inoculated plants under stress conditions to find their role in stress alleviation. To study the effects of PGPRs (MNCN1, LLCG23 and FZB42) on wheat root morphology in the presence of salt stress an automated Rhizoscanner, equipped with WinRHIZO software offered by Regent Instruments Co. was used to calculate different morphological root parameters, i.e., root length, surface area, volume, diameter and number of root tips for each wheat plant. Three wheat plants were randomly selected from each treatment in triplicate after 9 days’ post-inoculation for Rhizoscanning and the average values were calculated for the above parameters to see the effects of each PGPR on wheat root morphology under salt stress.

### 4.8. Determination of Photosynthetic Potential of Plants under Salt Conditions

Wheat plants grown at 25 °C under different NaCl concentrations were used for analysing photosynthesis rate and stomatal conductance performance by taking readings from five leaves of each plant using an open IRGA LI-COR 6400 XT portable photosynthesis system (LI6400, Li-Cor Inc., Lincoln, NE, USA). Net photosynthetic rate (Pn) and stomatal conductance (gs) were taken under saturated light conditions at a photosynthetic photon flux density of 1000 µmoL photons m^–2^ s^–1^ and 380 mol^–1^ CO_2_ concentrations.

### 4.9. Quantification of Plant Stress Response Parameters and Their Expression Profiling

The plant induces a response to salt stress in the presence of different stress responsive parameters. The wheat plants were studied for these stress responsive parameters under salt stress conditions. The treated and control plants were grown in the greenhouse in a controlled environment and analysed for proline accumulation, lipid peroxidation and abscisic acid (ABA). Seven days’ wheat seedlings were transplanted to pots inoculated with selected PGPRs grown under different salt stress conditions. The non-inoculated wheat seedlings were used as control. The leaves were randomly harvested in triplicates from each salt treatment after 7 days.

#### 4.9.1. Abscisic Acid (ABA) Determination

To quantify ABA in salt stress treated wheat plants, 0.1 g of each sample was ground into a fine powder with pestle and mortar in the presence of liquid nitrogen followed by transferring to 2 mL autoclaved tubes and stored at −80 °C to proceed later. The ground samples were then homogenised in 80% methanol and kept at 4 °C overnight in a stirring machine at 300 rpm. The samples were then centrifuged at 14,000 rpm at 4 °C followed by the collection of supernatants of each sample in new 2 mL tubes. The supernatants were dried through vacuum evaporation for 4 h at room temperature. The extract of each sample was then dissolved in 80% and subjected to UPLC for ABA detection and Quantification [66].

#### 4.9.2. Quantification of Proline in Wheat Plants under Salt Stress

Proline overproduction in plants is an important stress response indicator for study in different environmental stresses [67]. Fresh wheat leaves samples (0.1 g) grown under salt conditions and untreated as control were taken at 7 days and homogenised completely in 70% ethanol [68]. Briefly, in a 2:1 ratio, the ethanolic extract and a solution containing 60% acetic acid and 20% ethanol were mixed to form a homogenised leaf mixture of each sample. The volume of 100 µL reaction mixture of each sample was taken in a 1.5 mL test tube, heated up to 95 °C for 20 min followed by cooling at room temperature and the absorbance was measured at 520 nm in a microplate reader (Spectrum max plus; Molecular Devices, Sunnyvale, CA, USA).

#### 4.9.3. Estimation of MDA Level (Lipid Peroxidation) in Salt-Treated Wheat Plants

Lipid peroxidation is estimated in terms of malondialdehyde (MDA) levels studied in plants under stress conditions. Briefly, fresh leaves (0.1 g) of PGPRs-inoculated wheat plants were grown under salt conditions and untreated as a control was taken and homogenised in 0.1% (*w*/*v*) TCA 500 µL solution. The mixture was then centrifuged at 4 °C and 13,000 rpm. The supernatant of each treatment was then mixed with 1.5 mL of 0.5% TBA solution and heated to 95 °C in a water bath for 25 min. To stop further reactions, the mixture was placed on ice for 5 min. The absorbance of the reaction was recorded at 532 nm and 600 nm by using a microplate reader (Spectrum max plus; Molecular Devices, Sunnyvale, CA, USA).

#### 4.9.4. Expression Analysis of Growth and Salt Stress Related Genes in Wheat Plant

To find the expression pattern of the above stress responsive genes, i.e., *P5CS* for proline, *4-HNE* for MDA and *ABARE* for ABA and phytohormones genes (*ARF*, *ERF*, *expA1*, *CKX2*) and genes responsible for salt resistance (*DREB2*, *MYB*, *HKT1*, and *WRKY17)* was carried out through RT q-PCR in wheat plants grown under salt stress conditions (Appendix A). The selected gene sequences were taken from NCB1, followed by designing primers through the PrimerQuest tool. The RNA was extracted from PGPRs inoculated wheat plants grown under salt stress conditions at 7 days using the TRizol method [69]. The RNA of each sample was used for cDNA synthesis according to the protocol given with the cDNA kit. The cDNA of each sample was used in RT-qPCR to find out each gene expression profile. RT-qPCR was performed to find the expression profile of these genes in PGPRs inoculated or uninoculated wheat plant through Quant Studio Real-Time Thermocycler (Thermo Fisher Scientific, San Jose, CA, USA).

### 4.10. Statistical Analysis

All the in vitro and in planta experiments in this study were conducted completely randomised and repeated thrice. The data were expressed in ±SD (standard deviations) of three replicates (*n* = 3). One-way analysis of variance (ANOVA) was used to statistically evaluate the experimental data by using Statistix software ver. 8.1. The least significant difference test (Tukey HSD) was used to observe the significance among the treatment means at the probability level of *p* ≤ 0.05 [70].

## 5. Conclusions

The significance of the current research is to explore the genetic features of potential halophilic bacterial strains, NMCN1 and LLCG23, that make them able to perform efficient metabolic and physiological functions under high saline conditions. The ability of these bacteria to alleviate salt stress in wheat plants and promote plant growth through modulation of phytohormones and salt resistance genes. These beneficial plant microbe interactions achieved more attention for further research to use them efficiently in making promising bio-fertilisers or biopesticides for the purpose of obtaining better agricultural crops’ production in an extreme saline environment.

## Figures and Tables

**Figure 1 plants-11-02769-f001:**
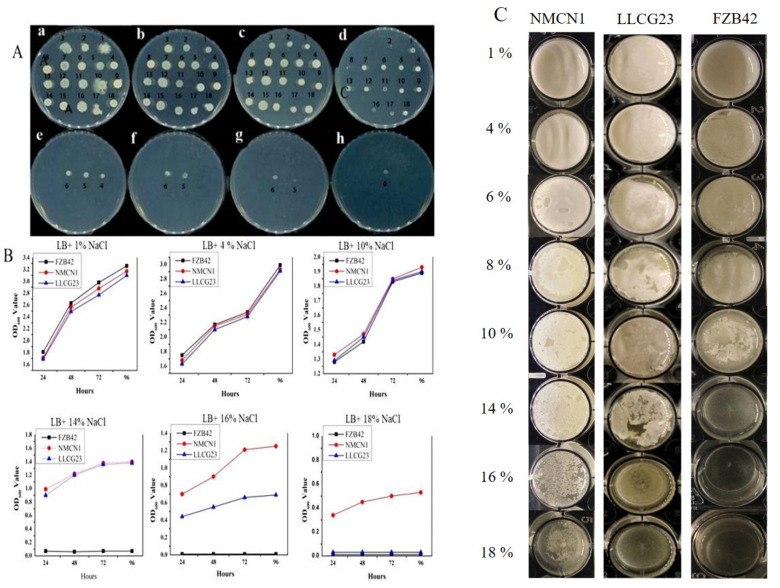
*Bacillus* spp. strains’ growth and biofilm formation under different saline conditions. (**A**) The visual representation of strains’ growth on solid LB media (a–h) with different NaCl concentrations (1–18%). Each number indicates strain: 1. FZB42, 2. B168, 3. EZ-1509, 4. CJCL2, 5. LLCG23, 6. NMCN1, 7. SYST2, 8. LNXM65, 9. TS1, 10. LSSC3, 11. GBSW11, 12. LNXM37, 13. LLTC93, 14. GBAC46, 15. NMCC4, 16. LNXM10, 17. LLTC96, 18. GBWC18. (**B**) Shows the graphical presentation of optical density at different time intervals calculated at (OD_600_) of high halotolerant strains (MNCN1 and LLCG23) as compared to FZB42 grown in Liquid LB media with different salt concentrations. (**C**) The biofilm formation potential of selected strains grown under different NaCl concentrations (1–18%) for 5 days at 37 °C. The experiment was repeated thrice with similar results.

**Figure 2 plants-11-02769-f002:**
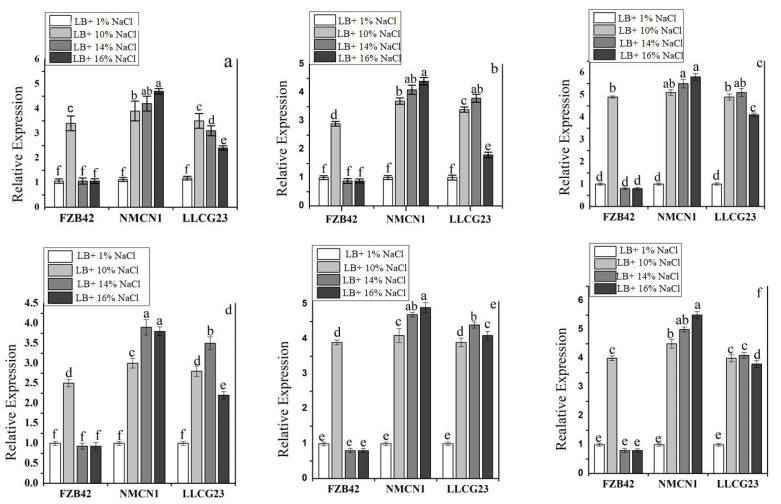
Expression analysis of predicted genes in *Bacillus* strains FZB42, NMCN1 and LLCG23 grown in different NaCl concentrations (1%, 10%, 14% and 16%). The genes analysed were: *DegU* (**a**), *OstB* (**b**), *OhrR* (**c**), *ComA* (**d**), *SodA* (**e**) *and OpuAC* (**f**). The error bars on the columns indicate the standard deviation of the mean (*n* = 3). Letters above the columns represent significant differences between treatments at *p* ≤ 0.05. The expression studies were repeated thrice with three replicates for each treatment.

**Figure 3 plants-11-02769-f003:**
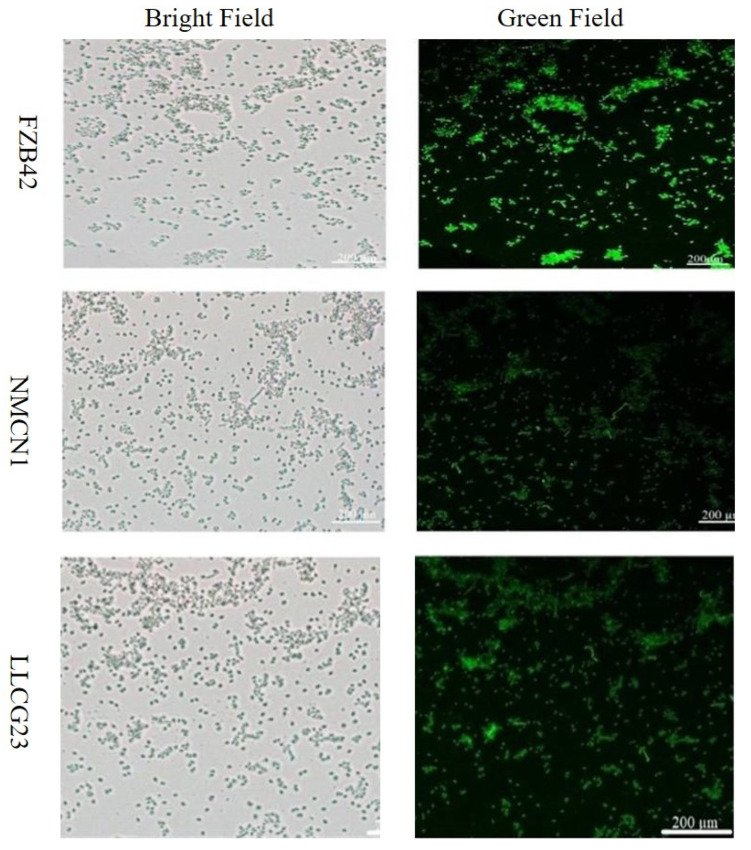
The intensity of green fluorescence indicates the level of Reactive Oxygen Species (ROS) in selected *Bacillus* strains grown at 37 °C in saline condition for 4 days. The less intense fluorescence observed in NMCN1 and LLCG23 strains describes lower ROS production in their cells under salt stress while the high intensive green fluorescence in FZB42 strain displays more ROS production under saline condition. The experiment was repeated thrice similar results.

**Figure 4 plants-11-02769-f004:**
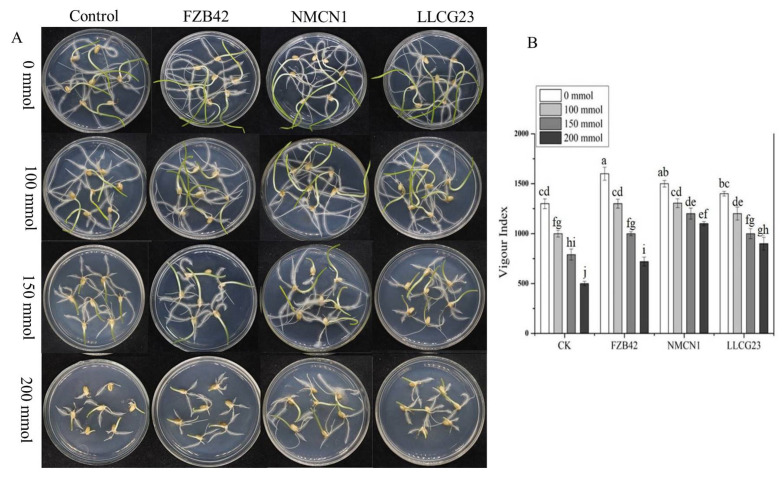
Wheat seedlings inoculated with *Bacillus* strains FZB42, NMCN1 and LLCG23 were grown for 7 days at 25 °C under different NaCl concentrations (0 mmol–200 mmol). (**A**) The visual representation showing the effect of selected strains on wheat growth promotion under different saline conditions. (**B**) The calculation of seedlings’ length and percent germination is represented in graph as vigour index for wheat seeds treated with different *Bacillus* strains as compared with control (seeds without bacteria). The bars on the column represent mean standard deviation (*n* = 3). The small letters on the bars indicate significant difference among the treatments calculated through Tukey’s HSD test at *p* ≤ 0.05 and the experiment was repeated thrice with three replicates for each treatment.

**Figure 5 plants-11-02769-f005:**
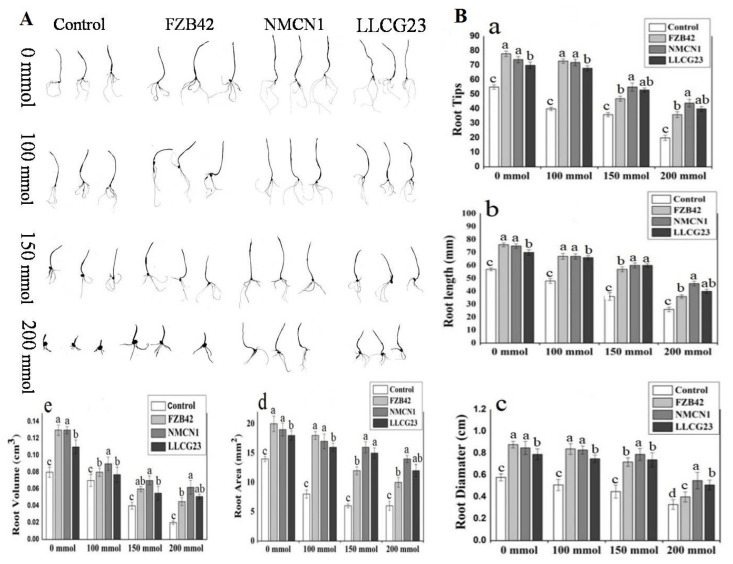
Rhizoscanning of wheat seedlings grown under different saline conditions (0 mmol–200 mmol NaCl) at 25 °C for 7 days insulated with Bacillus spp. strains. (**A**) Rhizoscanning in image form of wheat seedlings grown under different saline conditions at 25 °C. (**B**) Morphological root parameters such as (**a**) Root Tips, (**b**) Root Length (mm), (**c**) Root Diameter (cm), (**d**) Root Area (mm^2^), (**e**) Root Volume (cm^3^). The small letters on the bars indicate significant difference among the treatments calculated through Tukey’s HSD test at *p* ≤ 0.05 and the experiment was repeated thrice with three replicates for each treatment.

**Figure 6 plants-11-02769-f006:**
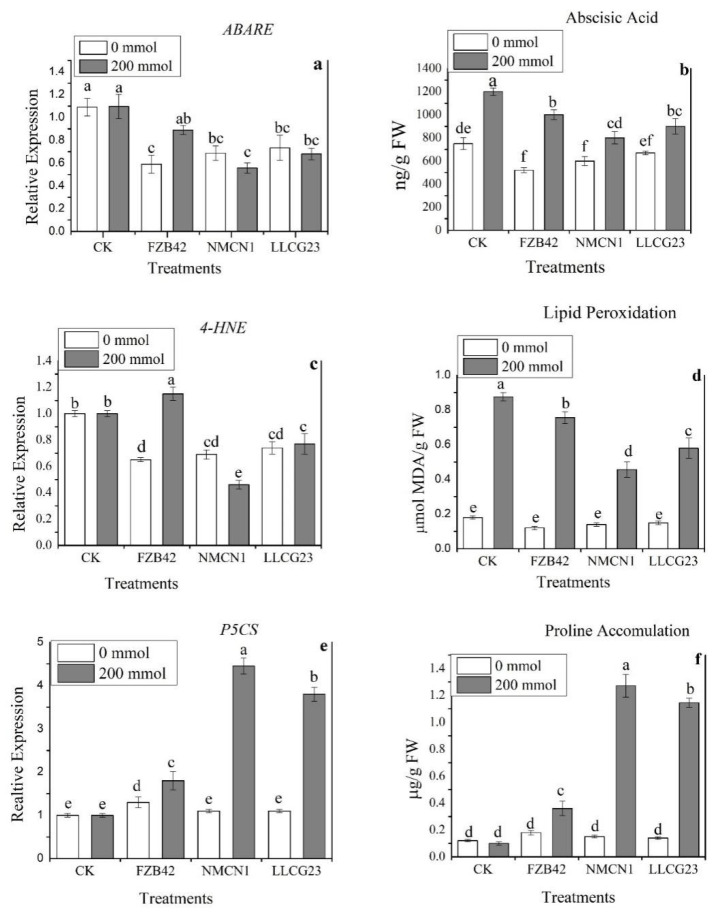
Relative expression and quantification of Abscisic acid, Proline and Lipid peroxidisation. (**a**) Expression profiling of abscisic acid (ABA) encoding gene *ABARE*, (**b**) ABA quantification through UPLC analysis in wheat plants under salt stress (200 mmol NaCl), (**c**) expression analysis of MDA encoding gene (*4-HNE*), (**d**) quantification of MDA content through spectrophotometric method in wheat plants grown under saline condition, (**e**) expression analysis of proline encoding gene *P5CS,* (**f**) spectrophotometric quantification of proline. The error bars on the columns represent the standard deviation of the mean (*n* = 3). The small letters were used to show significant difference among the treatments calculated through Tukey’s HSD test at *p* ≤ 0.05. The experiment was repeated thrice with three replicates for each treatment.

**Figure 7 plants-11-02769-f007:**
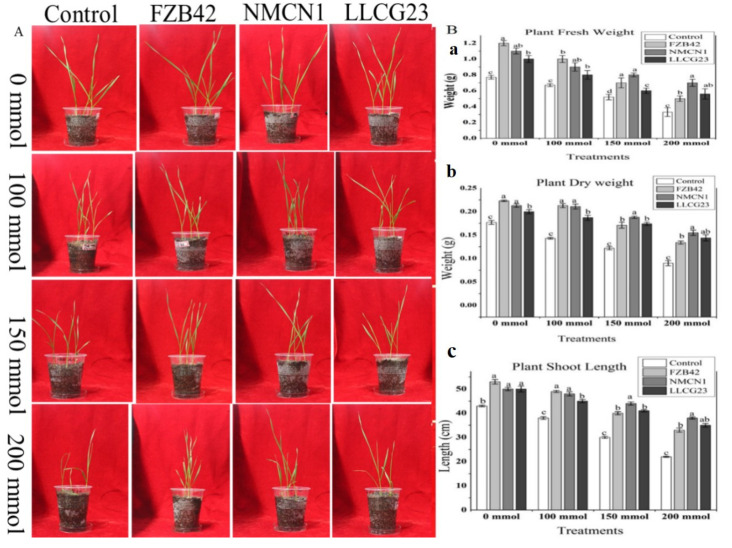
Effect of *Bacillus* spp. strains (FZB42, NMCN1 and LLCG23) on wheat plants grown under different salt conditions (0 mmol–200 mmol). (**A**) Visual representation showing the effect of the selected strains on wheat plants under saline conditions; (**B**) graphical detail of different plant parameters, i.e., (**a**) fresh weight of wheat plants inoculated with *Bacillus* strains grown in different salt stress conditions; (**b**) plants dry weight; (**c**) plant total shoot length. Standard deviations of mean (*n* = 3) were used to indicate error bars on the columns. The small letters were used to show significant difference among the treatments calculated through Tukey’s HSD test at *p* ≤ 0.05. The experiment was repeated thrice with three replicates for each treatment.

**Figure 8 plants-11-02769-f008:**
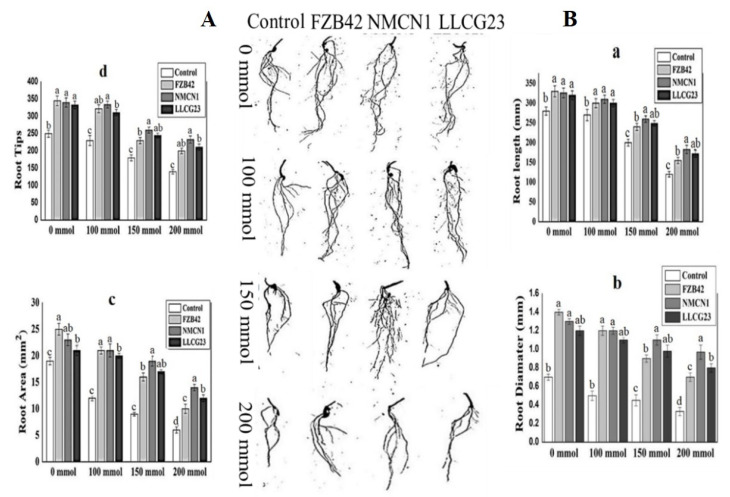
Wheat roots rhizoscanning image under different saline conditions (0 mmol–200 mmol) for 10 days treated with Bacillus strains (NMCN1, FZB42 and LLCG23) in a greenhouse experiment. (**A**) Visually gives the information regarding the effect of Bacillus strains on root morphology. The effect of selected strains on root morphological parameters (**B**) i.e., (**a**) wheat root total root length (mm) post inoculation of Bacillus strains grown under different salt stress conditions, (**b**) roots diameter (mm), (**c**) root total surface area (cm^2^), (**d**) number of root tips for each treatment. The error bars shown on the graphs represent the mean standard deviation of each treatment (*n* = 3). The small letters were used to show significant difference among the treatments calculated through Tukey’s HSD test at *p* ≤ 0.05. The experiment was repeated thrice with three replicates for each treatment.

**Figure 9 plants-11-02769-f009:**
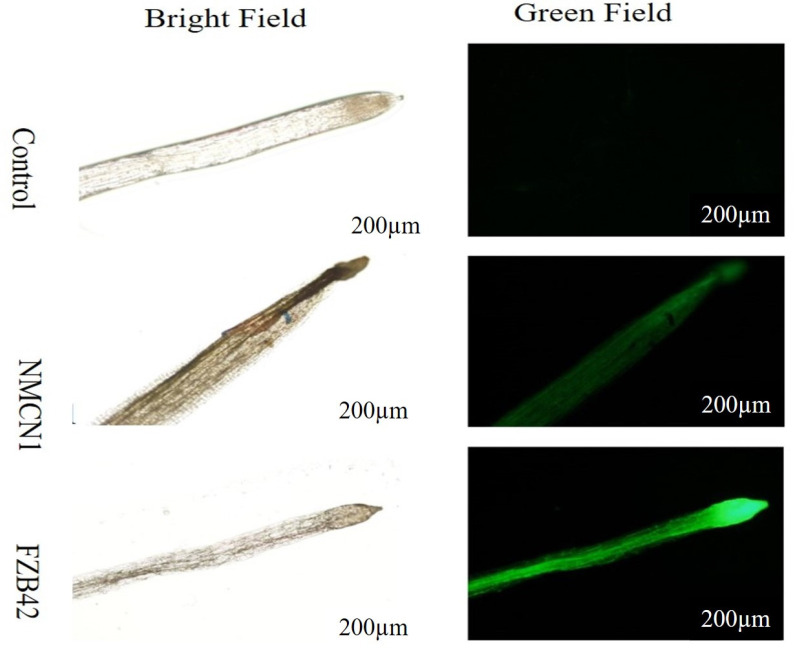
ROS reduction in wheat roots inoculated with *Bacillus* strains (NMCN1 and FZB42). The halophilic strains NMCN1 were found to highly reduced ROS in wheat roots and thus alleviate the risk of cell damage in roots. The experiment was repeated thrice with three replicates.

**Figure 10 plants-11-02769-f010:**
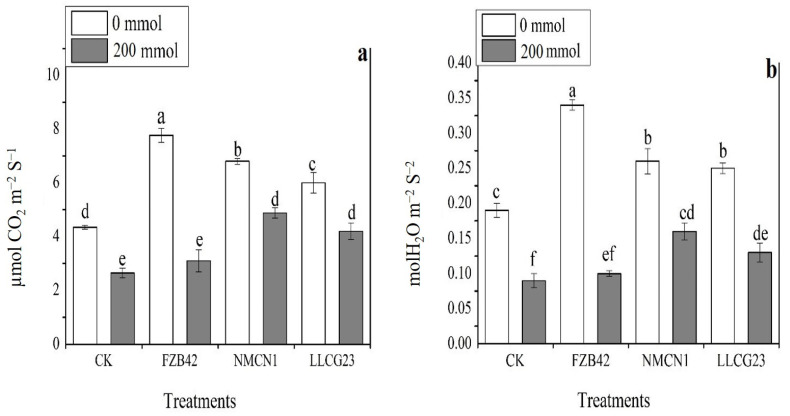
Effect of bacterial strains on photosynthesis rate and stomatal conductance regulation under salt stress. (**a**) Photosynthesis regulation of wheat plants inoculated with *Bacillus* strains under saline conditions; (**b**) *Bacillus* strains effects on stomatal conductance of wheat plants under different saline environment. The error bars represent the mean standard deviation of each treatment (*n* = 3). The small letters were used to show significant difference among the treatments calculated through Tukey’s HSD test at *p* ≤ 0.05. The experiment was performed in triplicate with similar results.

**Figure 11 plants-11-02769-f011:**
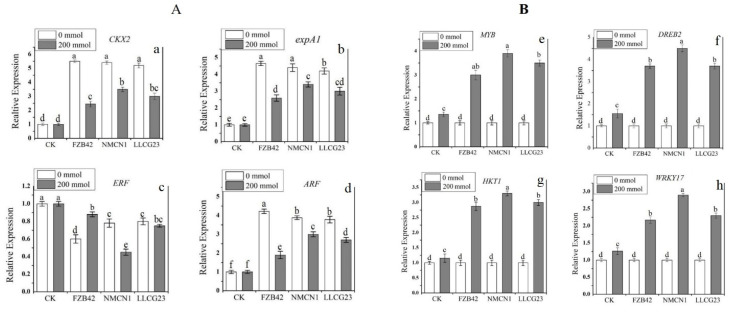
Effect of *Bacillus* strains on the expression of growth-promoting and salt-related genes in wheat plants grown under salt stress conditions (200 mmol). (**A**) Shows the expression profile of growth promoting genes (**a**–**d**) in wheat plants. (**B**) Indicates the expression analysis of salt-related genes (**e**–**h**) in wheat plants inoculated with *Bacillus* strains under saline condition (200 mmol). The error bars on the graphs indicate the standard deviation of the mean (*n* = 3). The small letters were used to show significant difference among the treatments calculated through Tukey’s HSD test at *p* ≤ 0.05. The experiment has repeated three times with three replicates for each treatment.

## Data Availability

Not applicable.

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
