# Peer review of "Salt Tolerant Bacillus Strains Improve Plant Growth Traits and Regulation of Phytohormones in Wheat under Salinity Stress"

_plants, 2022, doi:10.3390/plants11202769_

Round 1

Reviewer 1 Report

The manuscript entitled “Salt tolerant Bacillus strains improves plant growth traits and regulation of phytohormones in wheat under salinity stress” was evaluated. He proposes that the soil salinity is a major constrain adversely affecting most important agricultural crops including wheat world-wide. However, this research evaluated the genetic potential of high halophilic Bacillus strains, isolated from rhizosphere in the extreme environment of Qinghai-Tibetan plateau region of China, for reducing salt stress in wheat plants. In general, the manuscript is well written, but needs minor revision in order to be accepted Plants. Below are some considerations made.

1.      Abstract / Authors must include more details on results (may be in percentage);

2.      Introduction / Hypothesis can be improved with recent literature;

3.      Materials and methods / How were chosen Na+ concentrations and Bacillus strains?

4.      Materials and methods / More details on data analysis.

5.      Results / Subtitles can be shortened.

6.      Discussion / Authors must elaborate a sentence with a broad approach to the results obtained in this research.

7.      Discussion / What genes are involved in salt tolerance?

8.      Discussion / Authors must include more studies available in literature linked to salt effects on root system.

https://doi.org/10.1111/plb.13176

9.      Discussion / Authors must include more studies available in literature on oxidative stress and antioxidant responses generated during salt stress.

https://doi.org/10.1007/s00344-021-10481-5

https://doi.org/10.1007/s00344-021-10481-5

10.   Conclusion / In relation to antioxidant mechanism, authors must indicate the enzymes.

11.   Tables / Tables were well elaborated.

12.   Figures / Figures were well elaborated.

Author Response

Dear Editor,

Thank you very much for your attention and reviewers’ appreciations on our manuscript (Manuscript ID: plants-1955224). We highly acknowledge reviewers’ comments and suggestions, which were valuable in improving the quality of our manuscript. The detailed point-by-point responses to the referees’ comments are provided herewith. The manuscript has been revised accordingly using the “Track Changes” function in MS Word and a marked-up version of the manuscript is uploaded.

We sincerely hope that our manuscript will be finally acceptable for publication. Thank you very much for all your help and looking forward to your response.

Yours sincerely,

Prof. Qin Gu

Nanjing Agricultural University, Nanjing 210095, China.

E-mail: [email protected] (Q.G).

The manuscript entitled “Salt tolerant Bacillus strains improves plant growth traits and regulation of phytohormones in wheat under salinity stress” was evaluated. He proposes that the soil salinity is a major constrain adversely affecting most important agricultural crops including wheat world-wide. However, this research evaluated the genetic potential of high halophilic Bacillus strains, isolated from rhizosphere in the extreme environment of Qinghai-Tibetan plateau region of China, for reducing salt stress in wheat plants. In general, the manuscript is well written, but needs minor revision in order to be accepted Plants. Below are some considerations made.

 Response; Thank you very much for your helpful suggestions and valuable input in our research manuscript. We also very much appreciate the comments/suggestions made by referees. According to the suggestions, whole manuscript has been revised carefully. We also incorporated most of the suggestions during our revision. A point-by-point response is provided below. The revisions are highlighted in the main text with red color and track changes.

Abstract / Authors must include more details on results (may be in percentage).

Response: Thanks, the increase in growth parameters of wheat i.e. fresh weight, dry weight, shoot length and root length are mentioned in the abstract of the MS with percent value according with your suggestion.

Introduction / Hypothesis can be improved with recent literature.

Response: Thanks, the introduction/ hypothesis has been improved with recent literature. 

Materials and methods / How were chosen Na+ concentrations and Bacillus strains?

Response: Thanks, the Bacillus strains (NMC1 and LLCG23) were observed to grow under high saline conditions, followed by previous study (Ali et. al., 2022, Wu Xiaohui et. al., 2022). Then we were further to evaluated for their potential to alleviate the adverse effect of salt stress on wheat plants.

Ali, Q.; Ayaz, M.; Mu, G.; Hussain, A.; Yuanyuan, Q.; Yu, C.; Xu, Y.; Manghwar, H.; Gu, Q.; Wu, H.; et al. Revealing Plant Growth-Promoting Mechanisms of Bacillus Strains in Elevating Rice Growth and Its Interaction with Salt Stress. Front. Plant Sci. 2022, 13, 1–17, doi:10.3389/fpls.2022.994902.

  1. Wu, Y. Fan, R. Wang, Q. Zhao, Q. Ali, H. Wu, Q. Gu, R. Borriss, Y. Xie, X. Gao, Bacillus halotolerans KKD1 induces physiological, metabolic and molecular reprogramming in wheat under saline condition, (2022) 1–17. https://doi.org/10.3389/fpls.2022.978066.

Materials and methods / More details on data analysis.

Response: Thanks, we have explained more detail in material and methods / data analysis. 

Results / Subtitles can be shortened.

Response: Thanks, the subtitles of the results has been shortened.    

Discussion / Authors must elaborate a sentence with a broad approach to the results obtained in this research.

Response: Thanks, A sentence has been added in the discussion, which summarize the mains points of the results in the current research work. 

Discussion / What genes are involved in salt tolerance?

Response: Thanks, the genes names SodA, OpuAC, OPuD, OstB, ComA and DegU in Bacillus strains are thought to be involved in abiotic stresses (Salt and cold) directly or indirectly according to the previous study by Zubair et., 2019.

Discussion / Authors must include more studies available in literature linked to salt effects on root system.

https://doi.org/10.1111/plb.13176

Response:  Thanks, the discussion has been improved with recent literature and the paper has been cited.

Discussion / Authors must include more studies available in literature on oxidative stress and antioxidant responses generated during salt stress.

https://doi.org/10.1007/s00344-021-10481-5

https://doi.org/10.1007/s00344-021-10481-5

Response: Thanks, the mentioned papers has been cited in the MS providing more knowledge on oxidative stress and antioxidant responses generated during salt stress.

Conclusion / In relation to antioxidant mechanism, authors must indicate the enzymes.

Response: Thanks for your valuable suggestion. The current study mainly focused on the potential of Bacillus strains to combat with high saline conditions and their potential to regulated phytohormones and salt resistant genes in wheat. The antioxidant assay will be performed in the near future in another project in which we will study different mechanism and regulation of antioxidant enzymes for ROS scavenging by Bacillus strains in wheat plants under salt stress.

Tables / Tables were well elaborated.

Response: Thanks, we have checked the tables again and modified.

Figures / Figures were well elaborated.

Response: Thanks, the figures has been checked and well elaborated. 

Reviewer 2 Report

The research article “Salt tolerant Bacillus strains improves plant growth traits and regulation of phytohormones in wheat under salinity stress” having a interesting results with respect to various physiological processes and salt stresses. The authors have highlighted the role and regulation of bacillus strain in wheat plant growth under salt stress. Manuscript is well written. The manuscript could be accepted pertaining to certain modifications as follows:

Numbering of subtitle is not uniform.

Q. Antioxidants enzymes assay experiments are missing in MS. It would be better to add information about antioxidant enzyme activity under salt stress.

Line No. 16: Delete “most important”

Line No. 17: Expand PGPR

Line No. 558: Is common primer used for PCR and Real time PCR? The predicted genes were amplified through PCR and detected on gel electrophoresis.

Line No. 657: How much bacterial were added in each pot. Concentration is missing. When the OD600 reached to 1.0 the cell cultures of selected bacteria were 658 added to each pot.

Line No. 726: Primer information is missing for the gene DREB2, MYB, HKT1, and WRKY17.

Fig 7 A. Not clear. Add other good pixal figure.

Author Response

Dear Editor,

Thank you very much for your attention and reviewers’ appreciations on our manuscript (Manuscript ID: plants-1955224). We highly acknowledge reviewers’ comments and suggestions, which were valuable in improving the quality of our manuscript. The detailed point-by-point responses to the referees’ comments are provided herewith. The manuscript has been revised accordingly using the “Track Changes” function in MS Word and a marked-up version of the manuscript is uploaded.

We sincerely hope that our manuscript will be finally acceptable for publication. Thank you very much for all your help and looking forward to your response.

Yours sincerely,

Prof. Qin Gu

Nanjing Agricultural University, Nanjing 210095, China.

E-mail: [email protected] (Q.G).

The research article “Salt tolerant Bacillus strains improves plant growth traits and regulation of phytohormones in wheat under salinity stress” having a interesting results with respect to various physiological processes and salt stresses. The authors have highlighted the role and regulation of bacillus strain in wheat plant growth under salt stress. Manuscript is well written. The manuscript could be accepted pertaining to certain modifications as follows:

Response; Thank you very much for your helpful suggestions and valuable input in our research manuscript. We also very much appreciate the comments/suggestions made by referees. According to the suggestions, whole manuscript has been revised carefully. We also incorporated most of the suggestions during our revision. A point-by-point response is provided below. The revisions are highlighted in the main text with red color and track changes.

Numbering of subtitle is not uniform.

Response: Thanks, we have changed the numbering of subtitles in the MS.

Antioxidants enzymes assay experiments are missing in MS. It would be better to add information about antioxidant enzyme activity under salt stress.

Response: Thanks, the current study mainly focused on the potential of Bacillus strains to combat with high saline conditions and their potential to regulated phytohormones and salt resistant genes in wheat. The antioxidant assay will be performed in the near future in another project in which we will study different mechanism and regulation of antioxidant enzymes for ROS scavenging by Bacillus strains in wheat plants under salt stress.

Line No. 16: Delete “most important”

Response: Thanks, the word “most important” is deleted.

Line No. 17: Expand PGPR

Response: Thanks, the word PGPR is expanded in the MS.

Line No. 558: Is common primer used for PCR and Real time PCR? The predicted genes were amplified through PCR and detected on gel electrophoresis.

Response: Thanks, we have used two set of primers i.e. one for conventional PCR to amplify and detect the predicted salt related genes as mentioned in the manuscript and another set of primers were used for qPCR to find the expression of each gene under different salt conditions.

Line No. 657: How much bacterial were added in each pot. Concentration is missing. When the OD600 reached to 1.0 the cell cultures of selected bacteria were 658 added to each pot.

Response: Many thanks for your comment. We have added the information related to bacteria concentration in MS.

Line No. 726: Primer information is missing for the gene DREB2, MYB, HKT1, and WRKY17.

Response: Thanks, the detail of primers for the selected genes is given in the supplementary material. 

Fig 7 A. Not clear. Add other good pixel figure.

Response: Thanks, the good pixel figure is added in the MS.

Round 2

Reviewer 2 Report

MS is revised by authors satisfactory as per given suggestions.. MS can be accepted in present form